# Effects of *Auricularia auricula* Polysaccharides on Gut Microbiota Composition in Type 2 Diabetic Mice

**DOI:** 10.3390/molecules27186061

**Published:** 2022-09-16

**Authors:** Nannan Liu, Mengyin Chen, Juanna Song, Yuanyuan Zhao, Pin Gong, Xuefeng Chen

**Affiliations:** 1College of Chemistry and Materials Science, Weinan Normal University, Weinan 714099, China; 2School of Food and Biological Engineering, Shaanxi University of Science and Technology, Xi’an 710021, China; 3College of Food Science and Engineering, Gansu Agricultural University, Lanzhou 730070, China

**Keywords:** *Auricularia auricula* polysaccharide (AAP), hypoglycemic activity, gut microbiota, antioxidant capacity

## Abstract

In previous studies, *Auricularia*
*auricula* polysaccharides (AAP) has been found to improve type 2 diabetes mellitus, but its mechanism remains unclear. In this study, we sought to demonstrate that AAP achieves remission by altering the gut microbiota in mice with type 2 diabetes. We successfully constructed a type 2 diabetes mellitus (T2DM) model induced by a high-fat diet (HFD) combined with streptozotocin (STZ), following which fasting blood glucose (FBG) levels and oral glucose tolerance test (OTGG) were observed to decrease significantly after 5 weeks of AAP intervention. Furthermore, AAP enhanced the activities of total superoxide dismutase (T-SOD), catalase (CAT), and glutathione peroxidase (GSH-Px), and reduced the content of malondialdehyde (MDA) to alleviate the oxidative stress injury. AAP-M (200 mg/kg/d) displayed the best improvement effect. Moreover, 16S rRNA results showed that AAP decreased the abundance of Firmicutes and increased that of Bacteroidetes. The abundance of beneficial genera such as *Faecalibaculum, Dubosiella*, *Alloprevotella, and* those belonging to the family *Lachnospiraceae* was increased due to the intake of AAP. AAP could reduced the abundance of *Desulfovibrio, Enterorhabdus*, and *Helicobacter*. In all, these results suggest that AAP can improve the disorders of glucose and lipid metabolism by regulating the structure of the gut microbiota.

## 1. Introduction

Diabetes, which is primarily characterized by high blood glucose, is a metabolic disorder syndrome of sugar, proteins, and fat caused by the dysfunction of pancreatic islets and insulin resistance. It is expected that the number of diabetics will reach 600 million in 2040, especially type 2 diabetes mellitus (T2DM), accounting for more than 90% of the total [1,2,3]. It is a typical chronic metabolic disease caused by unhealthy living habits such as diet, lifestyle, apart from genetics and environment. Several studies have reported a link between the disorder of gut microbiota and T2DM [4]. Gut microbiota and its metabolites usually maintain a relatively stable ecological balance with the host [5,6]. In addition, it plays an important role in disease prevention and health maintenance. Once disrupted, certain pathogenic bacteria can cause metabolic diseases such as hyperlipidemia, hypertension, diabetes, and colon cancer [7]. Previous reports have proved that diabetes can lead to increased abundance of certain opportunistic pathogenic bacteria [8], reduced relative abundance of probiotics such as *Bifidobacterium* and *Faecalibacterium prausnitzii*, and decreased amount of short-chain fatty acids (SCFAs) and increased amount of inflammatory factors [9,10]. Hence, an understanding of the pathogenesis of T2DM and the targets of its drug action have increasingly focused on the structure of the human or animal gut microbiota.

Different types of dietary fibers and polysaccharides are utilized by the gut microbiota to produce metabolites, which consequently affect the composition of gut microbiota and help maintain intestinal homeostasis. Chen [11] used 16S rRNA and 1H nuclear magnetic resonance (NMR) to demonstrate that *Ganoderma lucidum* polysaccharide (GLP) intervention reduced the abundance of harmful bacteria such as *Aerococcus, Ruminococcus, Corynebacterium*, and *Proteus* and improved the levels of *Blautia, Dehalobacterium, Parabacteroides*, and *Bacteroides* in T2DM mice. In addition, it restored the amino acid metabolism, glucose metabolism, and nucleic acid metabolism to produce anti-diabetes effects. Similarly, Li et al. [9] investigated demonstrated that tea polysaccharides maintained the diversity of the gut microbiota and restored the relative abundance of certain bacterial genera including *Lachnospira, Victivallis, Roseburia*, and *Fluviicola*, in T2DM mice. *Auricularia auricula* is an important member of edible and medicinal fungi, whose polysaccharides have several advantages including reducing sterols [12], antioxidant properties [13,14], lowering blood lipids, [15] and regulating immune function [16,17]. Previous studies have reported that AAP can improve insulin resistance, and blood lipid levels, and repair pancreatic islet liver cells. Furthermore, metabolomics has elucidated the metabolic pathway of AAP-improving T2DM [18]. However, the mechanism by which AAP improves diabetes needs to be further explored. In this study, the improvement of AAP in high blood glucose symptoms and antioxidant capacity both in vivo and in vitro were studied. In addition, 16S rRNA was used to evaluate the effects of AAP on alleviating diabetes by regulating the gut microbiota structure in diabetic mice.

## 2. Results 

### 2.1. Monosaccharide Composition and Molecular Weight of AAP

The AAP was decomposed into monosaccharides by acidolysis and analyzed using IC as shown in Figure 1. Compared with mixed standard monosaccharide components (Figure 1A), AAP was composed of fucose, galactose, glucose, mannose, and glucuronic acid in approximate mole percentages of 4.24%, 13%, 34.26%, 33.88%, and 6.39%, respectively (Figure 1B). 

A polymer, including a polysaccharide, is a mixed system of homologs with different molecular weights. Thus, the molecular weights of the polymers are distributed over a range, generally expressed as an average. Figure 2 shows the gel permeation chromatogram of AAP with a number–average molecular weight (Mn) of 2.67 × 10^6^ (Da), a weight–average molecular weight (Mw) of 3.05 × 10^6^ (Da), and a Z-average molecular weight (Mz) of 5.48 × 10^6^ (Da).

### 2.2. Antioxidant Activities of AAP In Vitro

DPPH has been widely used to determine the antioxidant capacity of biological samples, classified substances and food [19]. The ability of AAP to clear DPPH is shown in Figure 3A. As the concentration of AAP increased, the DPPH clearance rate showed an upward trend. When the concentration of AAP reached 1 mg/mL, the scavenging ability of DPPH radical reached 43.3%.

Hydroxyl free radical has a strong oxidation ability and can interact with a variety of molecules in the body to cause oxidative damage to sugars, amino acids, proteins, and lipids [20]. The scavenging effects of AAP (0–1 mg/mL) on hydroxyl radicals are shown in Figure 3B. Hydroxyl radical scavenging rate was dose-dependent; at an AAP concentration of 1 mg/mL, the clearance rate of AAP was 38.7%.

Figure 3C shows the result of reducing power of AAP. The increase in absorbance was positively correlated with its reducing power. In the range of 0–1 mg/mL, the absorbance increased slowly with an increase in its concentration, with a significant difference with VC.

### 2.3. Effects of AAP on Body Weight, FBG, and OTGG of Mice

The change in body weight is an important physiological indicator to measure the health status of animals. Figure 4A shows the trend of weight change of mice during the experiment. In the beginning, there was no significant difference in the body weight of mice between the groups (*p* > 0.05). After 4 weeks of HFD feeding, the average weight of mice in the MD, AAP-L, AAP-M, AAP-H, and PC groups increased by 32.7%, 30.8%, 28%, 32%, and 27.4%, respectively. After successful modeling, the weight of diabetic mice showed a downward trend. After 3 weeks of AAP and metformin intervention, the rapid weight loss trend of T2DM mice was effectively stopped and remained relatively stable until the end of the experiment. The final body weight of mice in the MD group was significantly lower than that in the NC group (*p* < 0.01). The AAP-M group displayed the best effect on body weight stabilization, and its final body weight was significantly higher than that of MD (*p* < 0.05).

Fasting and postprandial blood glucose elevation are major indicators to diagnose diabetes. The FBG levels of mice in each group were within the normal range at the initial stage of the experiment. After successful induction of T2DM by STZ, the average FBG of T2DM mice exceeded 11.1 mmol/L and was significantly higher than that of mice in the NC group (*p* < 0.01). After a 2-week intervention, T2DM mice other than those in the AAP-M group exhibited increased FBG levels, especially those in the MD group (FBG increased by 50% compared to the time when the modeling was successful). This could be attributed to the fact that STZ can continuously destroy the pancreatic islets β cells. However, the FBG levels in the AAP-M group decreased significantly (*p* < 0.01). After a 4-week intervention, the FBG levels of all protection groups decreased to different degrees. Finally, the FBG levels of the AAP groups were significantly lower than that of the MD group (*p* < 0.01). Overall, the AAP-M group displayed the most significant hypoglycemic effect (Figure 4B). 

OGTT plays a significant role in the diagnosis of diabetes, especially for early diabetes and light diabetes. According to Figure 4C, blood glucose levels of mice in different groups were maximized 30 min after the administration of oral glucose, especially in the MD group. The blood glucose levels of all mice in the MD group exceeded 33.3 mmol/L. 60 min after the administration of oral glucose, blood glucose levels of mice in all groups decreased. Among them, the blood glucose levels of mice in the NC group were controlled rapidly and recovered to normal at 120 min. STZ-induced damage of islet β cells leads to insufficient regulation of blood glucose levels of diabetic mice, thereby resulting in poor blood glucose regulation of mice in the MD group. The blood glucose levels of mice in the AAP group and the PC group 90 and 120 min after the administration of oral glucose were significantly lower than those in the MD group (*p* < 0.01). Additionally, blood glucose levels of mice in the AAP-M group were 14.17 ± 1.56 mmol/L at 120 min, which were significantly lower than those of mice in other groups (*p* < 0.01), indicating an optimized hypoglycemic effect.

### 2.4. Effects of AAP on Antioxidant Enzyme Activities and Liver Glycogen in Liver Tissue

Recently, numerous studies have implicated oxidative stress in the occurrence and development of diabetes [21,22]. Continuous high blood glucose levels can increase the active oxygen content of the body and impair the secretion of insulin [23,24]. Increased oxidative stress in the liver is related to the occurrence and development of diabetes [25]. Compared with the NC group, the activity of T-SOD in the MD group decreased significantly (*p* < 0.01). After intervention with AAP and metformin for 5 weeks, the activity of T-SOD increased significantly compared with the MD group (*p* < 0.01), and the activity of T-SOD among the groups was significantly different (*p* < 0.01) (Figure 5A). GSH-Px, which widely exists in organisms, is an important anti-oxidase. It can catalyze GSH to GSSG, thereby reducing peroxides to non-toxic hydroxyl compounds [26]. Compared with the NC, the activity of GSH-Px and CAT of the MD group decreased significantly (*p* < 0.01). After AAP and metformin intervention, the administration group exhibited significantly enhanced activity of GSH-Px and CAT compared with the MD group (*p* < 0.01) (Figure 5B,C). Among them, the AAP-M group displayed the best effect on improving CAT (32.5 ± 1.51 U/mg protein). The MDA content decreased significantly after AAP intervention (*p* < 0.01). There was no significant difference between different treatment groups at the level of *p* < 0.01; however, at the level of *p* < 0.05, the reduction effect of AAP-M was apparent, about 18.1% lower than that of the MD group for MDA (Figure 5D). Compared with the MD group, the liver glycogen content was significantly increased after metformin and AAP protection (*p* < 0.01) (Figure 5E). Thus, AAP can reduce the oxidative stress damage caused by diabetes by increasing the activity of anti-oxidases and inhibiting lipid peroxidation in vivo.

### 2.5. Effects of AAP on the Composition of Gut Microbiota in T2DM Mice

Although active polysaccharides in natural products cannot be digested and absorbed directly by the human body, it can be decomposed in the human body by gut microbiota to produce SCFAs such as acetic acid, propionic acid, and butyric acid, thereby reducing the intestinal pH value, regulating energy balance, and maintaining intestinal homeostasis [11,27,28]. Changes in the gut microbiota in the cecal contents of NC, MD, AAP-M, and PC groups were analyzed by 16S rRNA sequencing to reveal the effects of AAP on the gut microbiota of T2DM mice.

Based on the results of species annotation and the abundance information of OTUs, an unsupervised multivariate statistical method was used to analyze the difference in the composition of the microbial community among different groups using beta diversity. The results of weighted principal coordinate analysis (PCoA) showed that after STZ induction, diabetic mice (MD, AAP-M, PC) and mice in the NC group exhibited significant differences in the structure of the gut microbiota. After AAP-M and PC treatment, the gut microbiota underwent certain structural changes along the second principal component (PC2) and was separated from the samples of the MD group (Figure 6A). The cluster diagram of different groups (Figure 6B) shows differences between the gut microbiota species of diabetic and normal mice. The abundance of Firmicutes in the NC group was higher than that in the other three groups. In addition, the abundance of Bacteroidetes in diabetic mice improved to different degrees compared with that in the NC group. The MD group was the farthest from the NC group, such that the difference between the groups was the largest. The AAP-M group and PC group showed a certain distance from the NC group and the MD group in cluster analysis, indicating that the treatment with AAP and metformin altered the gut microbiota of mice in the T2DM model. However, there still existed certain differences in the bacterial community structure from normal mice.

Figure 6C shows that the top 10 microorganisms with abundance at the phylum level included Firmicutes, Bacteroidetes, followed by Actinobacteria and Desulfobacterota. The proportions of Firmicutes and Bacteroidetes in the NC group mice accounted for 52.45% and 37.15%, respectively. The proportions of Firmicutes and Bacteroidetes in the MD group were 31.78% and 47.63%, respectively. The contents of Firmicutes in the cecum of mice in the NC group were significantly higher than those in mice in other diabetes groups (*p* < 0.05). Although there was no significant difference in the abundance of Firmicutes and Bacteroidetes among MD, AAP-M, and PC groups, the AAP-M group showed a more pronounced increase in the abundance of Bacteroidetes. Compared with the NC group, the MD group exhibited significantly increased Desulfobacterota content, which decreased after AAP-M intervention, especially after PC intervention (Figure 6D). 

According to Figure 6E,F, the abundance of gut microbiota in mice of each group at the genus level was different. The dominant genera of the NC group were *Ileibacterium*, *Dubosiella*, and *Lachnospiraceae*. After STZ induction, the abundance of these three strains in the MD group significantly decreased (*p* < 0.05). Following AAP intervention, the increasing trend of *Dubosiella* was strengthened, and the abundance of *Lachnospiraceae* in the AAP-M mice was significantly higher than that in the MD group (*p* < 0.05). Bacteria belonging to the family *Lachnospiraceae* are obligate anaerobic bacteria that produce butyric acid with rich content in the gut of animals and participates in the metabolism of bile acid [29,30], whereas butyrate-producing bacteria are scarce in the gut of patients with T2DM [7,31]. Following metformin or AAP intervention, the *Lachnospiraceae* content increased significantly, especially in the AAP-M group, compared with the MD group (*p* < 0.05) (Figure 6F). Certain studies have proved that *Dubosiella* is the key bacterium genus for normal lipid metabolism, and its abundance is significantly reduced in patients with fatty liver injury or alcohol-induced liver injury caused by HFD. Moreover, improving its abundance can effectively alleviate the injury [32]. In this study, the *Dubosiella* content increased significantly following AAP intervention, compared to the MD group. Additionally, *Desulforibrio* showed a high abundance in the MD group, which was reversed after AAP intervention. *Faecalibaculum* is also one of the typical bacteria that generate SCFAs (butyrate), and it is positively correlated with blood lipid levels [33]. In this study, the abundance of *Faecalibaculum* decreased significantly in the MD group compared with that in the NC group (*p* < 0.05), whereas intake of both AAP and metformin promoted the growth of *Faecalibaculum*. Additionally, abundance of *Alloprevotella*, *Enterorhabdus*, and *Helicobacter* was significantly reversed following AAP intervention, compared with the MD group. In summary, AAP can regulate and improve the altered structure of gut microbiota caused by T2DM.

Student’s *t*-test was used to obtain the abundance heat map at the genus level (Figure 7). The differential microorganisms, whose abundance was in the top 35, are reflected by the color depth. The dominant gut microbiota in the NC group included *Ileibacterium*, *Dubosiella*, *Romboutsia*, *Faecalibaculum*, *Mucispirillum*, *Alistipes*, and *Blautia*. Compared with the NC group, however, the MD group mice showed a higher reduction in the abundance of the above bacteria. After AAP intervention, the abundance of certain dominant bacteria genera in the MD group, such as *Desulfovibrio*, *Helicobacter*, *Enterorhabdus*, *Odoribacter*, and *Colidextribacter* decreased to varying degrees or close to the level of the normal group. Ma also investigated that in the diabetes model, *Enterorhabdus* abundance increased significantly, whereas Wuyi rock tea intervention significantly reversed this trend [34]. Compared with the NC group, the MD group exhibited low contents of *Alloprevotella*, *Eubacterium*_*xylanophilum*, *Lachnospiraceae*, and *Candidatus_Saccharimonas*. After AAP intervention, their abundance was significantly reversed. In summary, AAP can improve the altered gut microbiota of diabetes mice.

LefSe (LDA effect size) analysis was performed on different groups of bacteria to screen biomarkers with statistical differences between the groups. Based on species classification and abundance matrix, different species between groups are shown in LefSe circular tree (Figure 8). The circle represents the level from the phylum to the genus (or species) from inside to outside, and the abundance size is positively correlated with the diameter of the small circle. The nodes of red, green, blue, and bright yellow, respectively. represent the microbial groups that play an important role and have the largest abundance in the AAP-M, MD, NC, and PC groups, whereas light yellow indicates no significant difference, thus laying a potential important microbial group for different groups.

## 3. Discussion

The results of this study suggest that AAP can alleviate the symptoms of T2DM induced by HFD combined with STZ. Compared with the MD group, AAP significantly stabilized the weight, reduced FBG levels, and improved OTGG levels. Moreover, AAP intervention significantly enhanced the activity of T-SOD, CAT, and GSH-Px, and significantly reduced the MDA content, and reduced STZ-induced oxidative stress damage of T2DM mice. Hepatic glycogen content showed an upward trend following AAP treatment, thus promoting the synthesis of hepatic glycogen and reducing the symptoms of high blood glucose. Among them, the AAP-M group showed the best effect. The 16S rRNA analysis of the NC, MD, AAP-M, and PC groups showed that the community structure of gut microbiota was significantly different between the mice in the NC group and diabetic mice. The AAP treatment improved the structure and abundance of gut microbiota of diabetic mice.

An imbalance in the gut microbiota is known to be closely related to obesity, insulin resistance, diabetes, hyperlipidemia, and other metabolic diseases. It could serve as a new target for the treatment and improvement of such metabolic diseases [4,35]. There are two dominant bacteria phyla in the human body, namely Firmicutes and Bacteroidetes. Relevant studies have demonstrated that the occurrence of obesity and diabetes is related to the changes in the abundance of these two bacteria phyla [36,37] although certain studies report different results. Certain studies have shown that the abundance of Bacteroidetes is lower and that of Firmicutes is relatively high in HFD, obese, or diabetes [28,38], or the ratio of Firmicutes and Bacteroidetes relative abundance (F:B) in the gut of obese mice is increased, which decreases after dietary or drug intervention [7]. However, other studies revealed that the abundance of Bacteroidetes in patients with T2DM was significantly higher than that in people with normal glucose tolerance or prediabetes patients [9,10,27,39]. In this study, the MD group mice showed a decreased proportion of Firmicutes and an increased proportion of Bacteroidetes in the intestine, which could be related to mouse weight. This is because, at the later stage of establishing the diabetes model, the MD group exhibited significantly reduced weight, leading to the increase in the abundance of Bacteroidetes. At the genus level, the abundance of *Ileibacterium*, *Dubosiella*, and *Lachnospiraceae* was higher in the NC group, which decreased significantly in the MD group (*p* < 0.05). After AAP intervention, the abundance of *Lachnospiraceae* was significantly higher than that in the MD group (*p* < 0.05). *Lachnospiraceae* are obligate anaerobic bacteria involved in butyric acid production and participate in the metabolism of bile acids. AAP was reported to affect the metabolism of bile acids [18], which could be related to *Lachnospiraceae*. A study on the structural differences in the gut microbiota between colon cancer mice and normal mice showed that the abundance of *Lachnospiraceae* in mice with colon cancer was decreased significantly compared with the control group [40]. Guo [41] et al. reported important protective functions of *Lachnospiraceae* and *Enterococcaceae* against radiation-induced damage by targeting metabolomics and 16S rRNA sequencing. Li [42] investigated the effect of mung bean peptides (MBPs) on prediabetes and gut microbiota imbalance caused by an HFD. Compared with the normal group, the relative abundance of *Lachnospiraceae_NK4A136*_group in the HFD group showed a downward trend, whereas the relative abundance of it in the HFD + mung soybean peptide (MBPs) group showed an upward trend. *Dubosiella* is also a key bacterial genus for normal lipid metabolism. In patients with fatty liver injury or alcohol-induced liver injury caused by an HFD, the abundance of it was significantly reduced [6,43]. Li et al. reported that HFD reduced the abundance of *Dubosiella*, and improving its abundance can effectively alleviate the injury [42]. In this study, AAP intervention improved the abundance of *Dubosiella*. The abundance of *Faecalibaculum* decreased significantly in the MD group whereas intake of AAP promoted the growth of *Faecalibaculum*. Si et al. [44] studied the regulatory mechanism of blueberry anthocyanin extract (BAE) on the composition of serum fatty acids and gut microbiota of HFD mice. The results showed that BAE intake promoted the growth of *Faecalibaculum* and enhanced its antioxidant capacity. Qu et al. reported that Q14 fermented milk significantly increased the abundance of SCFA-producing bacteria *Bacteroides* and *Faecalibaculum* in the intestine of diabetic mice [45]. In addition, the content of the *Desulfobacterota* phylum in the model group increased significantly, which is consistent with the findings of previous studies [31]. Huang [46] reported that the content of *Desulfobacterota* in the intestines of diabetes with retinopathy was significantly higher than that in healthy people and diabetic patients without retinopathy, suggesting that *Desulfobacterota* phylum could potentially affect the late complications of diabetes. This study also confirmed this point. After AAP-M intervention, the abundance of the *Desulfobacterota* phylum decreased, demonstrating that AAP reduced the content of intestinal harmful bacteria.

## 4. Materials and Methods

### 4.1. Materials and Chemicals

*Auricularia**auricula* was purchased from Xixiang County, Shaanxi Province, China. Total superoxide dismutase (T-SOD), liver glycogen, glutathione peroxidase (GSH-PX), malondialdehyde (MDA), and catalase (CAT) kits were purchased from the Nanjing Jianguo Institute of Biological Engineering. Ascorbic acid and streptozotocin (STZ, purity ≥ 98%) were obtained from Sigma-Aldrich. Metformin was produced by Shanghai Squibb Pharmaceutical Company. Other reagents were purchased from Shanghai Yuanye Biotechnology Co., Ltd. (Shanghai, China).

### 4.2. Preparation of Auricularia auricula Polysaccharide 

The fruiting bodies of *Auricularia*
*auricula* were dried at 50 °C and crushed, and subsequently sifted through a 60 mesh. *Auricularia*
*auricula* polysaccharides (AAP) were obtained through hot water extraction and alcohol precipitation. The powder form was mixed with distilled water at a ratio of 1:90 and extracted at 70 °C for 4 h. After centrifugation (8000 rpm, 10 min), the supernatant was concentrated to a quarter of its volume and precipitated with 4 volumes of 95% ethanol at 4 °C for 24 h. The sediment was obtained by centrifugation (5000 rpm, 5 min). The remaining ethanol was evaporated and the protein was removed with Sevag reagent (n-butanol:chloroform = 1:4). Finally, the AAP was obtained using freeze-drying. In this word, the yield of AAP was 8.17%, and the total sugar content was 71.4%.

### 4.3. Monosaccharide Composition and Molecular Weight of AAP

The polysaccharides were decomposed into monosaccharides by acidolysis. The monosaccharides were analyzed qualitatively and quantitatively by ion chromatography (IC). Next, 5 mg (± 0.05 mg) of AAP samples were accurately weighed and added to the prepared trifluoroacetic acid (TFA) solution (2 mol/L), heated at 121 °C for 2 h, and blow-dried with nitrogen. Methanol was added for cleaning, followed by blow-drying again. The samples were cleaned two to three times with methanol and dissolved in 1 mL of sterile water for testing. Each sample was analyzed by Thermo ICS5000 chromatography system (ICS5000+, Thermo Fisher Scientific, Massachusetts, USA) and performed on a Dionex™ Carbopac™ PA10 (250 × 4.0 mm, 10 µm) liquid chromatography column with a 20 µL of the sample. The mobile phase consisted of A (H_2_O) and B (100 mM NaOH), and the column temperature was 30 °C.

The molecular weights of AAP were determined by gel chromatography-differential indicator (Optilab T-rEX (Wyatt, CA, USA)–multi-angle laser light scattering system (Dawn Heleos Ⅱ, Wyatt Technology). AAP sample (10 mg) was dissolved in 1 mL of the mobile phase (0.1 m NaNO_3_) and centrifuged at 14,000 rpm for 10 min. The supernatant was filtered by a 0.22 µM membrane before sample loading. The test conditions were as follows: gel exclusion column (Ohpak SB-805 HQ) = 300 × 8 mm, gel exclusion column (Ohpak SB-804 HQ) = 300 × 8 mm, gel exclusion column (Ohpak SB-803 HQ) = 300 × 8 mm), column temperature = 45 °C, sample size = 100 µL, flow rate = 0.4 mL/min, λ = 663.7 nm.

### 4.4. Antioxidant Activity of AAP In Vitro 

#### 4.4.1. DPPH Radical Scavenging Assay

The DPPH radical scavenging ability of AAP was analyzed according to the previously reported methods [47]. Briefly, 2 mL of the DPPH-ethanol solution (0.10 mmol/L) was completely mixed with 2 mL of the polysaccharide solution with gradient concentration. The absorbance was measured at 517 nm after standing for 30 min at room temperature without light. Distilled water was used as a blank control and VC was used as a positive control. Each sample was repeated thrice, and the average value was taken. DPPH clearance was calculated as follows:(1)$$\mathrm{DPPH}\;\mathrm{scavenging}\;\mathrm{activity}\;\%=1-\frac{A_{1}-A_{0}}{A_{2}}\times 100\%$$
where A_1_ is the absorbance value of the sample solution at 517 nm, A_2_ is the absorbance value of anhydrous ethanol replaced sample as control, and A_0_ was measured using anhydrous ethanol to replace DPPH solution as the blank.

#### 4.4.2. Hydroxyl Radical Scavenging Ability

Salicylic acid colorimetry was used as previously reported with appropriate modifications [48]. Briefly, 2.0 mL of FeSO4 (6 mmol/L) solution and 2.0 mL of H_2_O_2_ (6 mmol/L) solution were mixed to generate OH via the Fenton reaction. Next, 2.0 mL of sample solutions of different concentrations and 2.0 mL of salicylic acid solution (6 mmol/L) were added and mixed evenly. The absorbance was measured at 510 nm after standing at 37 °C for 30 min. Vitamin C was used as the positive control, each sample was measured thrice and averaged. The hydroxyl radical (·OH) radical-scavenging effect was calculated using the following formula: (2)$$\;\left(\cdot \mathrm{OH}\right)\;\mathrm{scavenging}\;\mathrm{effect}\;\left(\%\right)=1-\frac{A_{1}-A_{0}}{A_{2}}\times 100\%$$
where A_1_ is the absorbance of the sample, and A_0_ is the absorbance measured by distilled water instead of the sample. A_2_ is the absorbance measured by anhydrous ethanol instead of the sample.

#### 4.4.3. Determination of Total Reducing Capacity [49]

In total, 2.5 mL of sample solutions of different concentrations, 2.5 mL of a sodium phosphate buffer (0.2 M, pH 6.6), and 2.5 mL of 1% potassium ferricyanide solution were accurately measured in a 25 mL colorimetric tube and incubated at 50 °C for 20 min, followed by rapid cooling. Next, 2.5 mL of 10% trichloroacetic acid was added. Next, the mixture was centrifuged (3500 rpm, 10 min). Afterward, 100 µL of the supernatant was mixed with 20 µL of 1% ferric chloride and 100 µL of double steam water, and the absorbance of the mixture was recorded at 700 nm. Distilled water was used as the blank control and VC was used as the positive control. Each experiment was repeated thrice. The absorption value directly reflected the reducing power, and a greater absorption value corresponded to a stronger reducing power.

### 4.5. Animal Test

Sixty C57BL/6J male mice (specific pathogen-free [SPF] level, weight = 20 ± 2 g, age = 6–7 weeks) were purchased from Xi’an Ensiweier Biotechnology Co., Ltd. All mice were raised in SPF facilities at 22 ± 2 °C, humidity of 50% ± 10%, and light/dark alternation for 12 h. Mice were caged for every five mice, and water and feed were made freely available. After 7 days of feeding, 10 mice were selected as the normal control group (normal control, NC) and fed with normal diet. The remaining mice were used to establish the T2DM model, that is, they were fed with high-fat diet (17.5% lard, 12% sucrose, 1% whole milk powder, 13% casein, 2% calcium bicarbonate, and 43.7% standard pellet feed) for 4 weeks, followed by a 12 h fasting, intraperitoneal injection of STZ (35 mg/kg b.w for four days consecutively). STZ was dissolved in 0.1 M cold citric acid buffer (pH = 4.2–4.4), whereas mice in the NC group were administered with 0.1 M sodium citrate buffer for 4 days consecutively, followed by a 12 h fasting. The tail clipping measurements indicated that fasting blood glucose (FBG) varied from 11.1 to 33.3 mmol/L, which was considered as a successful model. The mice with successful modeling were randomly divided into five groups, with 10 mice in each group; these were namely, the model group (MD), the AAP-low group (AAP-L), the AAP-medium group (AAP-M), the AAP-high group (AAP-H), and the positive control group (PC). Mice in each group were treated by gavage once a day for 5 consecutive weeks. During this period, mice could eat and drink freely. The gavage treatment and feed administration of each group are shown in Table 1.

Before anatomy, mice were made to fast for 12 h, blood was collected from the orbit, and placed at room temperature for 2 h. After centrifugation (3000 rpm, 5 min), the serum was obtained and stored at −80 °C. After dissection, the liver was taken, rinsed with normal saline, and stored at −80 °C. The cecal contents of mice in each group were collected, stored in liquid nitrogen, and quickly transferred to a −80 °C ultra-low temperature refrigerator.

In this study, all animal treatments and experimental procedures followed the guidelines of the National Institutes of Health. All animal experimental procedures were reviewed and approved by the Institute of Basic and Translational Medicine of Xi’an Medical College.

### 4.6. Weight, FBG, and OTGG of Mice

After adaptive feeding, the changes in the weight of mice were monitored weekly. Specifically, the tail was cut to collect blood to determine the initial FBG of mice. Once the STZ-induced T2DM model was established, the FBG was measured again. In addition, FBG was measured weekly during the AAP intervention. The oral glucose tolerance test (OGTT) was performed following the method reported elsewhere [50]. Before anatomy, all mice were made to fast for 12 h, and the gavage glucose was 2.0 g/kg b.w. Blood was collected by tail clipping at 0, 30, 60, 90, and 120 min after gavage, and blood glucose levels of mice in different groups were determined. 

### 4.7. Measurement of Hepatic Glycogen and Anti-Oxidase Activities in Liver Tissues

The activity of hepatic glycogen, T-SOD, and CAT, the contents of GSH-Px and MDA in liver tissue homogenates of mice in different groups were measured using specific kits (Nanjing Jiancheng, Nanjing, China). 

### 4.8. Gut Microbiota Analysis

The total DNA was extracted from the cecum using the cetyltrimethylammonium bromide/sodium dodecyl sulfate (CTAB/SDS) method. The DNA concentration and purity were monitored on 1% agarose gels. According to the concentration, the DNA was diluted to l µg/µL using sterile water. The distinct regions of 16S rRNA (V3–V4) were amplified using specific primers, namely, forward primer: 341F (5’-CCTAYGGGRBGCASCAG-3’) and reverse primer 806R (5’-GGACTACNNGGGTATCTAAT-3’). All PCR reactions were performed with 15 µL of Phusion^®^ High-Fidelity PCR Master Mix (New England Biolabs) under the following thermal cycling steps: initial denaturation at 98 °C for 1 min, followed by 30 cycles of denaturation at 98 °C for 10 s, annealing at 50°C for 30 s, and elongation at 72 °C for 30 s, and finally holding at 72 °C for 5 min. The PCR products were detected on a 2% agarose gel and purified using the Qiagen Gel Extraction Kit (Qiagen, Dusseldorf, Germany). Sequencing libraries were generated using the TruSeq^®^ DNA PCR-Free Sample Preparation Kit (Illumina, San Diego, CA, USA). Afterward, the purified amplicons were sequenced on an Illumina NovaSeq platform and 250 bp paired-end reads were generated. The sequences of original DNA fragments were merged and assembled using the FLASH software, and the high-quality data were performed under specific filtering conditions according to the QIIME software (V1.9.1). Sequences analysis were performed by Uparse software (Uparse v7.0.1001). According to 97% similarity, representative sequences of OTUs (operational taxonomic units, OTUs) were obtained for species annotation to obtain the corresponding species information and abundance distribution. Multiple sequence alignments were conducted using MUSCLE software (Version 3.8.31) to study the phylogenetic relationships of different OTUs. 

### 4.9. Statistical Analysis

All data were indicated as means ± standard deviations (SDs). One-way analysis of variance (ANOVA) according to SPSS 19.0 and Student’s *t*-test were used to investigate the significant differences in the results. Values of *p* < 0.05 and *p* < 0.01 denoted significant differences at different levels. Beta diversity were calculated by QIIME software (Version 1.9.1). R software (Version 2.15.3) was used for PcoA analysis, and to analyze the composition structure at the genus level and to cluster the top 35 genera in abundance and draw the heatmap. QIIME software (version 1.9.1) was conducted for Unweighted Pair-group Method with Arithmetic Means(UPGMA) Clustering.

## 5. Conclusions

The ameliorating effects of AAP on high-fat-binding STZ-induced T2DM were investigated. The results demonstrated that AAP significantly reduced the levels of FBG, stabilized the weight, improved the levels of OTGG, reduced liver gluconeogenesis, promoted the synthesis of hepatic glycogen, and improved the activity of anti-oxidases in the liver, reduced the oxidative stress damage in T2DM mice. AAP can regulate the structure and composition of the gut microbiota in diabetic mice, and increase the beneficial bacteria producing SCFAs, including *Lachnospiraceae*, *Faecalibaculum*, *Dubosiella*, and *Alloprevotella*, which play crucial roles in maintaining the intestinal ecological balance. Although research on gut microbiota is still in the initial stage, the results of this study provide theoretical support for the application of AAP in the prevention and adjuvant treatment of diabetes, as well as in the design and production of hypoglycemic functional foods with AAP as the functional factor.

## Figures and Tables

**Figure 1 molecules-27-06061-f001:**
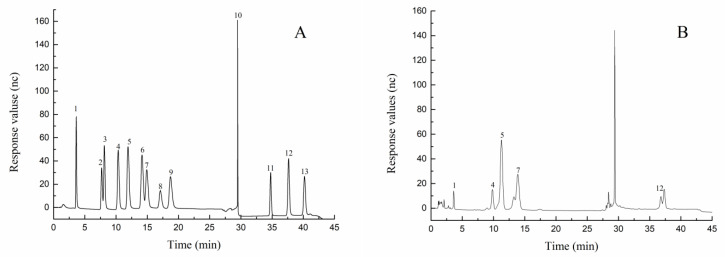
(**A**) Ion chromatography of mixed monosaccharides 1–13 represent (1) fucose, (2) rhamnose, (3) arabinose, (4) galactose, (5) glucose, (6) xylose, (7) mannose, (8) fructose, (9) ribose, (10) elution solvent, (11) galacturonic acid, (12) glucuronic acid, (13) mannuronic Acid. (**B**) Monosaccharide compositions of AAP. (1) fucose, (4) galactose, (5) glucose, (7) mannose, (12) glucuronic acid.

**Figure 2 molecules-27-06061-f002:**
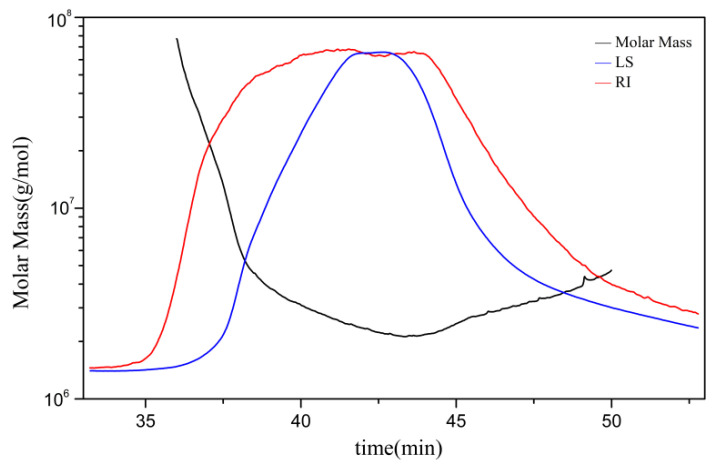
Gel permeation chromatogram of AAP.

**Figure 3 molecules-27-06061-f003:**
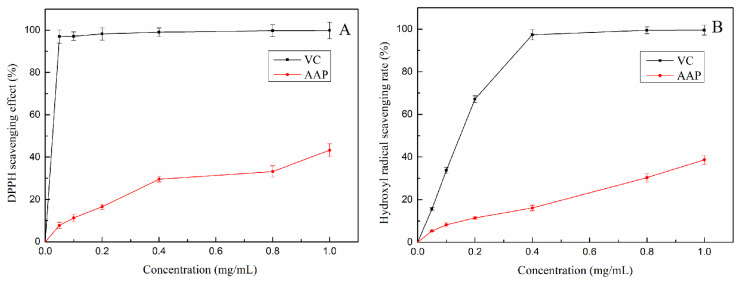
Comparison of antioxidant activity of AAP and VC of AAP in vitro (**A**) DPPH free radical scavenging ability; (**B**) hydroxyl radical scavenging ability; (**C**) reducing power.

**Figure 4 molecules-27-06061-f004:**
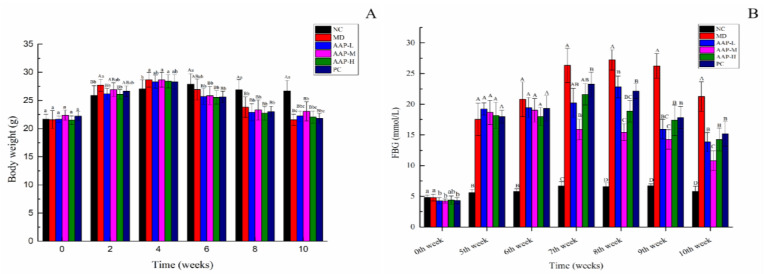
Effects of AAP on body weight, FBG, and OTGG of diabetic mice. (**A**) Body weight; (**B**) FBG; (**C**) OTGG. Note: The same letter indicates no significant difference between groups, and different letters indicate a significant difference between groups, A, B, C, D, *p* < 0.01; a, b, c, d, *p* < 0.05.

**Figure 5 molecules-27-06061-f005:**
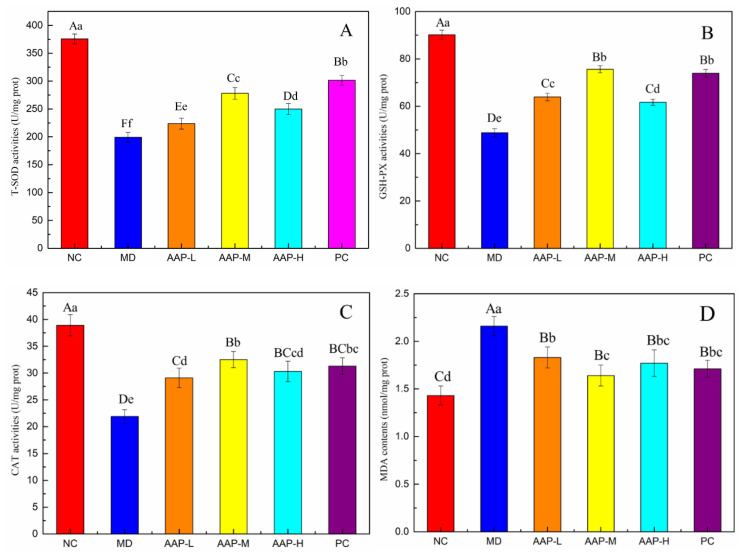
Effects of AAP on liver antioxidant capacity and hepatic glycogen content in mice. (**A**) T-SOD activities; (**B**) GSH-Px activities; (**C**) CAT activities; (**D**) MDA contents; (**E**) Hepatic glycogen content. Different letters indicate a significant difference between groups, A, B, C, D, *p* < 0.01; a, b, c, d, *p* < 0.05.

**Figure 6 molecules-27-06061-f006:**
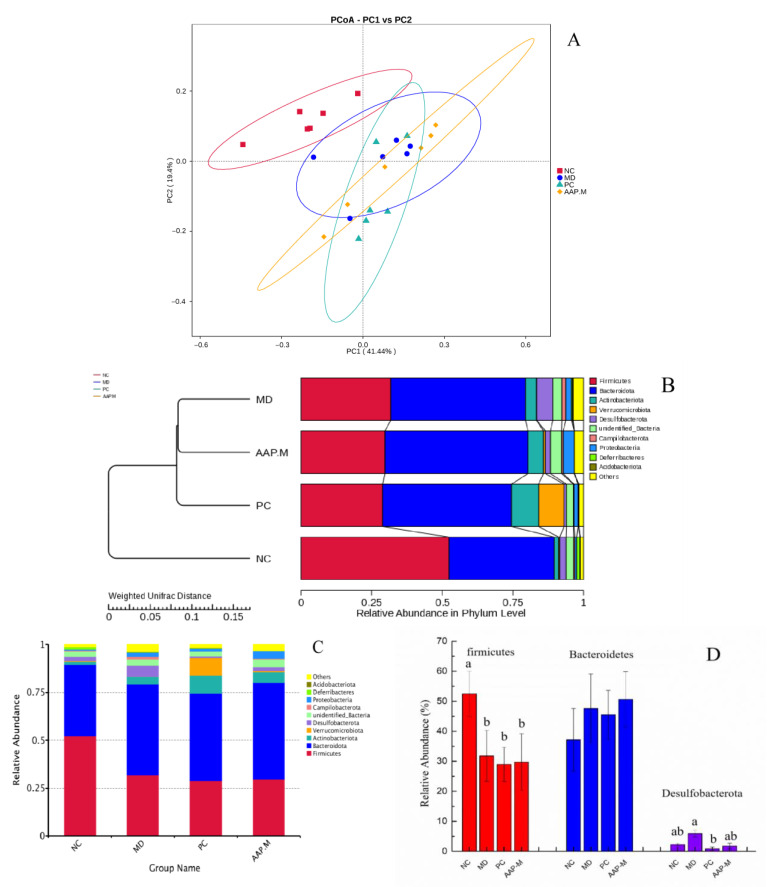
Differences in gut microbiota components in different groups of mice. (**A**) PcoA clustering analysis; (**B**) UPGMA clustering tree; (**C**) relative abundance of gut microbiota at the phylum level; (**D**) significant differences in the three phyla regulated by AAP; (**E**) relative abundance of gut microbiota at the genus level; and (**F**) relative abundance of nine genera among different groups. Note: Different letters indicate significant differences between groups, whereas the same letters indicate no significant differences. a, b, *p* < 0.05).

**Figure 7 molecules-27-06061-f007:**
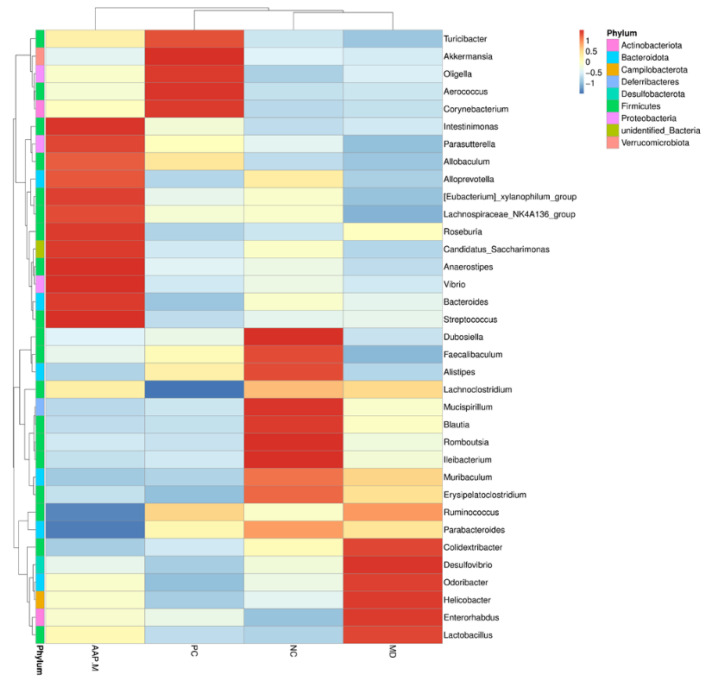
Heatmap of gut microbiota structure with significant changes at the genus level.

**Figure 8 molecules-27-06061-f008:**
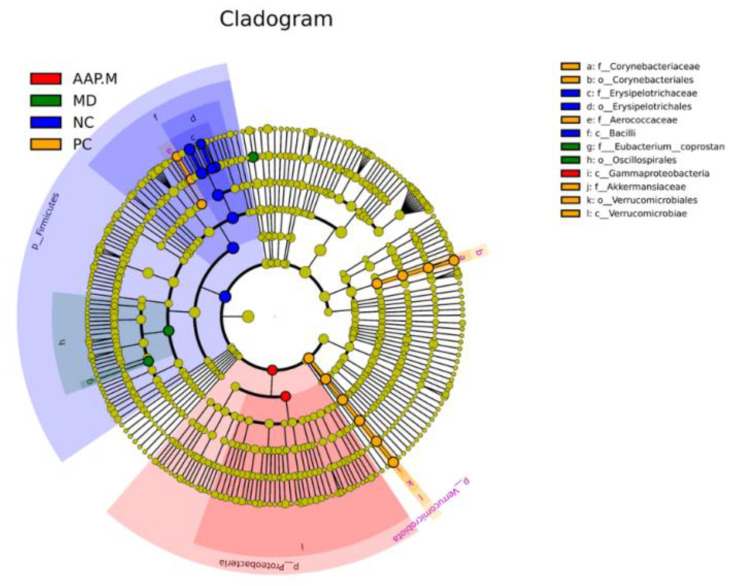
The circular tree of LefSe analyses.

**Table 1 molecules-27-06061-t001:** Intragastric treatment of animals.

Animal Groups	Intervention Dose	Feed
Normal control (NC)	Distilled water	Standard feed
T2DM model (MD)	Distilled water	High-fat feed
AAP-low (AAP-L)	100 mg/kg	High-fat feed
AAP-medium (AAP-M)	200 mg/kg	High-fat feed
AAP-high (AAP-H)	400 mg/kg	High-fat feed
Metformin positive control (PC)	200 mg/kg	High-fat feed

## Data Availability

Not applicable.

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
