# Peer review of "Effects of Auricularia auricula Polysaccharides on Gut Microbiota Composition in Type 2 Diabetic Mice"

_molecules, 2022, doi:10.3390/molecules27186061_

Round 1

Reviewer 1 Report

In this study, it demonstrated that Auricularia auricula polysaccharides (AAP) achieved remission by altering the gut microbiota in mice with type 2 diabetes. It can be accepted after revision.

1. AAP in this study in crude polysaccharides, what is ye yield and purity of the AAP?

2. Why did antioxidant activity in vitro experiments in this study?

3. Figure 2 should be improved.

Reviewer 2 Report

The current study was conducted to investigate the effect of Auricularia auricula polysaccharides (AAP) on the gut microbiota in a type 2 diabetes mellitus (T2DM) induced by induced by a high-fat diet (HFD) combined with streptozotocin (STZ). The results showed that AAP significantly stabilized the weight, reduced FBG levels, improved OTGG levels, and induced oxidative stress damage of T2DM mice. In addition, AAP significantly affected the bacterial community composition in the gut of T2DM through increasing the beneficial bacteria producing SCFAs, including Lachnospiraceae, Faecalibaculum, Dubosiella, and Alloprevotella.

Minor comments:

1. Section 2.9. Statistical analysis should have details of all analyses that were used. For examples, multivariate analysis, heatmap, and phylogenetic methods were not mentioned.

2. Line 211, 3. Results and Discussion, change to 3. Results

3. Section 3. Results should only include comments on the results, but this section had many interpretations and references which need to be removed or moved to section 4 (discussion).  

For example:

Line 217-225.

Line 306-309.

Line 325-327.

Line 369-373……..

So, the result section should be carefully revised and rewritten.
